# Associations of Broader Parental Factors with Children’s Happiness and Weight Status through Child Food Intake, Physical Activity, and Screen Time: A Longitudinal Modeling Analysis of South Korean Families

**DOI:** 10.3390/ijerph21020176

**Published:** 2024-02-03

**Authors:** Kay W. Kim, Jan L. Wallander, Bokyung Kim

**Affiliations:** 1School of Social Sciences, Humanities and Arts, University of California, Merced, CA 95343, USA; jwallander@ucmerced.edu; 2Simple Steps Community Connection, Palo Alto, CA 94303, USA

**Keywords:** play, stress, happiness, dietary intake, physical activity, weight status, parent, child

## Abstract

This study investigated how broader parental factors including parental happiness, parental play engagement, and parenting stress are related to Korean children’s happiness and weight status across three years via indirect pathways through the children’s energy-related behaviors of healthy and unhealthy food intake, physical activity, and screen time. Data from 1551 Korean parent pairs and 7-year-old children in the Panel Study on Korean Children were analyzed. A path analysis and gender-based multi-group analysis were conducted. Maternal happiness was negatively related to child screen time. Maternal play engagement showed positive concurrent associations with child healthy food intake and physical activity and negative associations with screen time. Maternal parenting stress was negatively related to child healthy eating. There was one significant finding related to fathers’ role on children’s energy-related behaviors, happiness, and weight status: the positive association between parental happiness and boys’ unhealthy food intake. Child screen time was positively related to child weight status and negatively to child happiness at each age. Broader maternal parenting factors can serve as a protective factor for childhood happiness and weight status in 7-to-9-year-olds through being associated with a reduction in child screen time.

## 1. Introduction

Childhood is a prime time to build healthy habits that nurture the foundation of a healthy and happy lifestyle [1]. Yet, numerous children adopt detrimental habits before they transition into adolescence [2]. These habits encompass an insufficient consumption of fruits and vegetables, overindulgence in energy-dense foods laden with sugars and fast food, reduced physical activity (PA), and excessive screen time [3]. Such energy-related behaviors (ERBs) may have broad effects on children’s health and happiness. Poor ERBs not only give rise to immediate health concerns, but also to obesity and overweight body mass index (BMI), which can often persist into adulthood [4]. Childhood obesity is a major public health concern as it has rapidly exacerbated in the past decades from 4% in 1975 to over 18% in 2016 worldwide [5]. South Korea (Korea, hereafter) is not an exception as the childhood obesity rate has increased from 9% in 2007 to 19% in 2021 [6].

Childhood happiness is a critical component of child well-being. The strong association between a close parent–child relationship and child happiness is well established [7]. Also, there is a general belief that PA is linked to happiness in children, while excessive screen time is associated with reduced happiness, often manifesting as higher levels of mental distress [8]. Likewise, a body of studies reports an association between the intake of fruit and vegetables and enhanced well-being in adults [9], but more research is clearly needed especially in the formative childhood period [10]. Of concern is that Korean children report comparatively low levels of happiness, placing them at the bottom of the 22 Organization for Economic Cooperation and Development countries [11]. Therefore, it is important to gain a deeper understanding of mechanisms that may contribute to childhood happiness, of which ERBs may be one set of candidates. 

### 1.1. Parental Correlates of Children’s Energy-Related Behaviors 

In childhood, at least prior to teenage years, parents are the primary influence on their children’s ERBs [12]. Much previous research has focused on specific strategies that parents use to influence their children’s ERBs, such as modeling exercise, restricting access to energy-dense snacks, and providing specific feedback on food choices [13]. However, recent studies have focused on broader approaches that parents use to provide parenting more generally that are not specifically targeting the children’s ERBs. Examples of such *broader parenting factors* are fostering cohesion in the family, applying the authoritative/democratic parenting style, and general monitoring of children [14]. Parents contribute to creating a nurturing emotional atmosphere in the home based on their own perceptions and experiences, particularly during children’s formative years [15], that are in part reflected by their approach to their children’s ERBs. For instance, a positive and supportive household environment can encourage health-promoting behaviors among children [16,17], while a stressful or negative environment can lead to unhealthy coping strategies such as emotional eating, overeating, or excessive screen time [18]. Understanding the role of broader parental factors can provide important insight into the potential barriers or facilitators to promoting healthy ERBs in children and can help inform interventions’ aims at improving the family environment and in turn children’s ERBs. Whereas a range of broader parental factors can be hypothesized to play a role in children’s ERBs, the current study focused on three: parental happiness, parental play engagement, and parenting stress.

### 1.2. Parental Happiness

Parental subjective happiness warrants attention for its influence on children’s development and health as happier parents possess better psychological resources, which enable the use of their emotional and social capabilities to provide a positive and warm home environment [19]. Because happiness is related to better physical and psychosocial well-being [20], happier parents tend to sustain the child’s development and emotional security [21]. Despite its link to numerous positive outcomes in children, research into links between parental subjective happiness and children’s ERBs is lacking thus far. We hypothesize that parental happiness is associated with more healthy and less unhealthy ERBs in their children.

### 1.3. Parental Play Engagement

Another important positive broader parenting factor is parents’ engagement in play with their children. Play involves a range of instinctive activities for recreational pleasure and enjoyment [22] and is such an essential foundation of children’s life that the United Nations High Commissioner for Human Rights declared it as every child’s right [23]. Play in a variety of scenarios, such as pretend play, role play, and building with blocks, etc., serves as a catalyst for the development of a wide spectrum of children’s competencies including executive functioning, cognitive aptitude, and effective communication abilities [24,25]. The active participation of capable parents in play with their children, encompassing both the quantity of time spent together and the quality of the interactions, enhances the transition to more intricate and advanced development of abilities such as planning, organization, and the use of verbal instructions [24]. Additionally, parents’ active involvement in play with their children fosters secure and nurturing relationships [26], which provides the underpinning for further positive development into and through adolescence. Despite the central role of parent–child play in children’s development, to our knowledge, no study has examined its association with children’s ERB. We hypothesize that it can provide a practical avenue for enhancing the parent–child relationship, which we argue is foundational for influencing child ERBs [14].

### 1.4. Parenting Stress 

While parents’ subjective happiness and play engagement are positive aspects of the home environment, parenting stress refers to the negative emotional experiences of strain, worry, anxiety, or depression that parents may experience specific to fulfilling their challenging parental responsibilities [27]. Parenting stress exerts a profound influence on parental interactions and parenting methods, and consequently, the overall well-being and development of children [28]. Moreover, parenting stress can significantly affect not only the parents’ own health behaviors but also those of their children [29]. However, the research findings regarding the connection between parenting stress and children’s ERBs have yielded mixed results. For example, children whose parents experience elevated parenting stress were inclined to consume fewer vegetables, spend more time on screens, and engage in less PA [29]. Conversely, other research showed that there was a positive association between parenting stress and unhealthy parental practices but not with children’s unhealthy dietary habits [30]. Consequently, there is a need for further research into the relationship between parenting stress and children’s ERBs. We hypothesize that parenting stress will be associated with fewer positive and more negative ERBs among children. 

### 1.5. Differences between Maternal and Paternal Factors on Daughters and Sons

Much of the research illuminating the role of broader parenting factors on children’s development has focused on mothers [31,32]. This may be due to gendered parenting practices where mothers spend substantially more time on average with children as the main caregiver and take on more household responsibilities especially related to ERBs [33]. In contrast, fathers tend to engage in relatively more play and leisure activities with their children [34,35]. Even so, mothers still participate more in children’s play and leisure activities than fathers in several countries including Korea and the United States [36,37]. In turn, these discrepancies in their parenting roles and related experiences may result in differential parenting experiences among fathers and mothers [38]. While it is widely acknowledged that mothers experience higher stress levels and lower overall happiness compared to fathers [38], our comprehension of the diverse pathways connecting the distinct influences of fathers and mothers on children’s ERBs, happiness, and weight status remains rudimentary. Especially in a culture where mothers are seen as the primary child caregiver, such as Korea, we expect that the mothers’ influence on children’s ERB will be stronger than that of fathers.

Moreover, some research confirms that parents’ parenting style varies depending on the child’s gender [39]. Parents use different socialization approaches and show different interaction patterns with boys compared to girls [40]. There are also gender differences in ERBs in that boys consume more calories [41] and engage in more PA and screen time than girls [42]. Also, in Korea [43], as in many Western countries [44], male adolescents have reported higher happiness than females, yet obesity rates among Korean boys are higher than girls [45]. These findings suggest that both the gender of the parent and the child are important in understanding parental influences on child development [40], and more research is needed to establish the complex links in the parent–child dyad in different gender combinations. Thus, it will be important to test for similarities and differences between fathers and mothers in the role of broader parental factors for daughters’ and sons’ ERBs, happiness, and weight status to better inform intervention efforts to improve childhood happiness and to prevent childhood obesity. 

### 1.6. Research Hypotheses

The overall aim of this research is to elucidate the role of ERBs in Korean children’s health and well-being, focusing on the indicators of weight status and happiness, as well as how broader parenting factors are associated with children’s ERBs. The focus is on the elementary school years from ages 7 to 9, a period mostly prior to when children are increasingly exposed to influences outside those of their parents. We propose a model depicted in Figure 1, which identifies relationships of parental happiness, parental play engagement, and parenting stress with children’s ERBs, and in turn, between their ERBs and happiness and weight status across two years. Specifically, the following hypotheses based on this model will be tested using a structural equation modeling (SEM) path analysis.

**H1.** 
*Parental happiness and play engagement are positively related to child healthy ERBs (healthy eating and PA) and negatively to child unhealthy ERBs (unhealthy eating and screen time).*


**H2.** 
*Parenting stress is negatively related to child healthy ERBs and positively to unhealthy ERBs.*


**H3.** 
*Child healthy ERBs are associated positively with child happiness and negatively with child weight status cross-sectionally as well as longitudinally one and two years later.*


**H4.** 
*Child unhealthy ERBs are associated negatively with child happiness and positively with child weight status cross-sectionally as well as longitudinally one and two years later.*


**H5.** 
*The relationships hypothesized in H1 and H2 will be stronger for mothers compared to fathers.*


Moreover, applying SEM, we also conduct a multi-group analysis to explore differences between boys and girls, as specified in Figure 1. 

## 2. Materials and Methods

### 2.1. Data Source and Participants

Data are from Wave 8 through 10 (child ages 7–9) of the publicly available data set from the Panel Study on Korean Children (PSKC) conducted by the Korean Institute of Child Care and Education [46]. The PSKC is a prospective longitudinal survey of a representative national cohort sample of children born between April and July 2008 and their parents. It was designed to collect comprehensive data on the characteristics of children, parents, families, and local communities as well as the effectiveness of childcare policies in Korea. The first wave of PSKC enrolling 2150 families was conducted in 2008, and follow-up surveys have been performed annually and are still ongoing. 

Participants are ethnically highly homogenous, reflecting Korean society. Only mothers who could communicate in Korean were invited at the time of child delivery. Also, mothers and newborns with serious health issues were excluded. The number of responding families was 1598 in Wave 8 in 2016 (child age 7), 1525 in Wave 9 in 2017 (child age 8), and 1484 in Wave 10 in 2018 (child age 9). The retention rate over the seven years from Wave 1 to Wave 8 was 75.3% [47]. This places this study well within the range of retention rates reported in other national longitudinal cohort studies. For example, the National Longitudinal Survey of Children and Youth in Canada retained 60% [48] whereas growing up in New Zealand retained 85% [49], in both cases being over eight years. Moreover, the current sample at Wave 8 is representative of the enrolled sample on major demographic characteristics (see Section 3.1, Table 1). For example, fathers’ and mothers’ education level retained identical medians and highly similar distributions over the ensuing period, other than the mothers with a 4-year college education, increasing by 3% seven years later (details available from authors). 

Written informed consent was obtained from each adult participant at the time of recruitment. The main caregiver provided consent for their child’s participation in this study. Methodological details of the PSKC have been reported elsewhere [50].

### 2.2. Measures

Measures of parental happiness, parental play engagement, and parenting stress, and child ERBs, were administered at child age 7 and those of child happiness and child weight status are in the form of a BMI percentile at ages 7, 8, and 9. 

#### 2.2.1. Parental Happiness

Parental happiness was assessed with the Subjective Happiness Scale (SHS) [51], which uses 7-point response scales tailored to the four questions: (1) “In general, I consider myself” rated from “not a very happy person” to “a very happy person”, (2) “Compared with most of my peers, I consider myself” rated from “less happy” to “happier”, (3) “Some people are generally very happy. They enjoy life regardless of what is going on, getting the most out of everything. To what extent does this characterization describe you?” rated from “not at all” to “a great deal”, and (4) “Some people are generally not very happy. Although they are not depressed, they never seem as happy as they might be. To what extent does this characterization describe you?” rated from “not at all” to “a great deal”. Mothers and fathers completed these items separately, yielding separate scores for each. The internal consistency reliability across items was α = .90 for mothers and .88 for fathers. Higher scores indicate a higher level of parental happiness. To test whether the items can support the measurement of parental happiness as a latent construct, individual items were subjected to a confirmatory factor analysis (CFA).

#### 2.2.2. Parental Play Engagement 

Parents indicated their level of play engagement by reporting the frequency of various parent–child play activities. The questionnaire was based on the Home Environment, Activities, and Cognitive Stimulation Questionnaire used in the Early Childhood Longitudinal Study Kindergarten Cohort [46]. The questionnaire consisted of 10 play activities, each rated on a 4-point scale from “never” to “every day”, including, for example, “I tell stories to my child”, “I do arts and crafts with my child”, “I talk about nature or do STEM project with my child”, and “I do block and puzzles with my child”. Mothers and fathers completed these items separately. The internal consistency reliability across items was α = .86 for maternal play engagement and .88 for paternal play engagement. Higher scores indicate a higher level of play engagement by a parent. These items were also subject to a CFA.

#### 2.2.3. Parenting Stress

Parenting stress was assessed with 11 questions from an instrument developed previously [52] and then revised for the PSKC during the pilot study in 2007. Questions were rated on a 5-point scale from “strongly disagree” to “strongly agree” and included, for example, “I am not sure if I could become a good parent”, “I am not sure if I could raise my child well”, “I feel sometimes that my child is behind his or her peers because I am not doing enough as a parent”, etc. Mothers and fathers completed these items separately. The internal consistency reliability across items was α = .90 for maternal stress and .89 for paternal stress. Higher scores indicate a higher level of parenting stress. These items were also subject to a CFA.

#### 2.2.4. Child Healthy Eating

The main caregiver (mostly mothers) was asked four questions addressing if the child (a) eats one serving of fruit or fruit juice per day; (b) eats vegetables, excluding Kimchi, at every meal; (c) eats lean meat/fish/bean/tofu, etc., at every meal; and (d) drinks two cups of milk/yogurt per day. The questions were developed by the PSKC and the Childhood Allergy Department at Asan Medical Center in Seoul, Korea. Responses were recorded on a 3-point scale including “very unlikely”, “moderately likely”, and “very likely”. A total composite score was calculated so that a higher score indicates more intake of healthy food items [53].

#### 2.2.5. Child Unhealthy Eating

Parents were asked three questions addressing if their child (a) eats fried food more than twice a week; (b) adds salt or soy sauce to the meals often (to make the meal salty); and (c) has ice cream, cake, or soda more than twice a week. The questions were developed by the PSKC and the Childhood Allergy Department at Asan Medical Center in Seoul, Korea. Responses were recorded on a 3-point scale including “very unlikely”, “moderately likely”, and “very likely”. A total composite score was calculated so that a higher score indicates more intake of unhealthy food items [49].

#### 2.2.6. Child Physical Activities 

Parents were asked how many minutes of the day a child usually spends time exercising, including Taekwondo, playing with balls, swimming, and free play outside in a park, play area, or yard of the house. This was reported separately for weekdays and weekends. The weekly total physical activity in minutes was calculated.

#### 2.2.7. Child Screen Time

Parents were asked how many minutes of the day a child usually spends time watching TV, a computer, and other screen-based devices and gaming activities with such devices. This was reported separately for weekdays and weekends. The weekly total screen time in hours was calculated. 

#### 2.2.8. Child Happiness

Child happiness was assessed using six items of a child self-report from an instrument developed previously [54] and then translated and revised by the PSKC for better understanding [55]. A survey researcher conducted in-person visits to the participants’ households. The children were orally asked questions about how they feel about their life and expressed their feelings using a face scale, which featured a range of four face images representing levels of agreement from “not very happy” to “very happy”. Questions included, “how do you feel when you think about your family?” and “what do you think of your current school?” The children indicated their responses by pointing to the picture that closely matched their feelings. The use of this type of response scale is common when assessing children, who may have difficulty with numerical or word anchors. Many common measures of child well-being and health employ such a response scale [56,57]. Moreover, systematic reviews have confirmed that psychometric data satisfactorily support the use of face response scales with children [56,58]. An overall score was calculated as an observed variable, with higher scores indicating greater happiness. The internal consistency reliability across items was α = .70 at age 7, .72 at age 8, and .74 at age 9. A previous study based on PSKC data using this measure confirmed its satisfactory internal consistency reliability and provides initial support for its construct validity [53,55]. 

#### 2.2.9. Child Weight Status 

Child’s weight status was represented by the BMI percentile score. Height and weight at ages 7, 8, and 9 were reported by the main caregiver and used to calculate the BMI percentile at each age, using the Korean Centers for Disease Control and Prevention gender- and age-specific charts [59]. A higher score indicates higher weight status.

### 2.3. Statistical Analysis 

IBM SPSS Statistics 28 was used for descriptive statistics and Mplus for an SEM analysis to test overall model fit and hypothesized associations. After the normality of all variables was checked, the square root transformation was applied to child PA to enhance normality. Missing data were computed under maximum likelihood estimation [60].

To assess the construct validity of the measurement models for latent constructs, a CFA was conducted separately for the items of maternal happiness, paternal happiness, maternal play engagement, paternal play engagement, maternal parenting stress, and paternal parenting stress. After ensuring the adequate fit for the measurement models, the SEM path analysis was conducted, reflecting the hypothesized model (see Figure 1) using observed scores to represent the latent variables. The highest education level between the mother and father and the number of people living in the household were added as control variables in the analyses of all paths. Child happiness and the child BMI percentile assessed at ages 7, 8, and 9 were set to covary with the previous assessment of the same variable.

In testing structural models, three goodness-of-fit indices were utilized to determine how well the model reproduced the characteristics of the observed data: the root mean square error of approximation (RMSEA), which should be less than 0.08 [61] and the comparative fit index (CFI) and Tucker Lewis Index (TLI), both of which should exceed 0.90 [62]. This was followed by a multiple-group SEM analysis to test differences in model parameters between boys and girls.

## 3. Results

### 3.1. Demographics 

Demographics for the sample are provided in Table 1. Parents on average are well educated as 73% of fathers and 71% of mothers had at least a 2-year college education. The average age at first assessment at Wave 8 was 40.31 for fathers, 37.91 for mothers, and 7.33 for children. The large majority of the families consisted of two parents and at least one child (88%). Descriptive statistics for the parental variables are shown in Table 2, and the child variables are reported in Table 3. Correlations among all study variables are reported in Appendix A.

**Table 1 ijerph-21-00176-t001:** Demographics.

Variable (N = 1551)	
Father’s age, Year (SD)	40.31 (3.95)
Mother’s age, Year (SD)	37.91 (3.72)
Child’s age, Month (SD)	87.91 (1.54)
Child’s gender (%)	
Boy	51.3
Girl	48.7
Father respondent’s educational level (%)	
8th grade or less	0.5
High school graduate and GED	26.2
Some college, or 2-year degree	20.2
4-year college graduate	42.3
More than a 4-year college degree	10.8
Mother respondent’s educational level (%)	
8th grade or less	0.4
High school graduate and GED	28.8
Some college, or 2-year degree	27.7
4-year college graduate	37.5
More than a 4-year college degree	5.6
Household composition (%)	
Parents and child/ren	88.4
Grandparents, parents, and child/ren	7.4
Parents, child/ren, and relative/s	0.8
Grandparents, parents, child/ren, and relative/s	3.3
Other	0.1

Note. SD = standard deviation.

### 3.2. Measurement Model

As shown in Table 4, the factor loadings for latent variables from the CFA indicated that observed variables loaded significantly onto their respective latent variables (maternal and paternal happiness, maternal and paternal play engagement with child, and maternal and paternal parenting stress). As shown in Table 4, all measurement models showed a close fit.

### 3.3. Path Analysis for the Total Sample 

Model fit was satisfactory (CFI = .98, TLI = .97, RMSEA: .03). Significant paths are detailed in Figure 2. Maternal happiness was negatively associated with child screen time, and maternal parenting stress was negatively related to child healthy eating. Maternal play engagement was positively associated with child healthy eating and PA and negatively with screen time. None of the paths from paternal variables to any of the child ERBs were significant. Out of the four child ERBs, only child screen time showed consistently positive association with child weight status and negative association with child happiness, measured at each of the three yearly waves. Child happiness was positively related from one year to the next, from age 7 to 8 and from age 8 to 9. Child weight status was likewise positively related from one year to the next.

### 3.4. Path Analysis for the Boys and Girls 

Model fit was satisfactory (CFI = .97, TLI = .97, RMSEA: .03) for the multi-group SEM. Significant paths for boys and girls are shown in Figure 3. Maternal happiness was negatively related to girls’ but not boys’ screen time. Paternal happiness was positively related to boys’ unhealthy eating. Maternal play engagement had a negative association with boys’ unhealthy eating and a positive association with boys’ and girls’ healthy eating and girls’ PA. Maternal parenting stress was negatively related to boys’ and girls’ healthy eating. Child screen time was negatively related to boys’ and girls’ happiness across three years, while child screen time was positively associated only with girls’ weight status concurrently and one year later. Child happiness was positively related from one year to the next within boys as well as girls, from age 7 to 8 and from age 8 to 9. Child weight status was likewise positively related from one year to the next for both boys and girls.

## 4. Discussion

We investigated the complex family dynamics interacting with Korean children’s ERBs and ultimately their happiness and weight status. Mainly, we found that it was the mothers who appear to exert more influence on children’s behaviors in the form of eating habits, PA, and screen time. Regardless of child gender, having a mother who engaged in more play was associated with children eating more healthy food, being more physically active, and spending less time in front of screens. Among child ERBs, it was the child screen time that had the strongest association with child happiness and weight status. As suggested, children who spent more time with screens were linked to lower ratings of happiness and higher weight status. There were few parental and child gender interactions. We found that higher maternal happiness was only related to girls’ spending less time with screens. Also, more maternal play engagement was only related to girls’ higher PA. The sole paternal factor that exhibited significance was the counterintuitive association between fathers’ higher happiness and more unhealthy eating in boys. 

### 4.1. Maternal Happiness and Children’s Screen Time

To our knowledge, this is the first study to test and report a negative association between mothers’ happiness and children’s screen time. Although we are not aware of any studies examining this link directly, a recent study conducted in Finland involving preschool-aged children found that parents reporting higher levels of happiness tended to have children who engaged in a greater number of healthy ERBs generally [63]. As happiness is rooted in better physical, psychological, and social well-being [20,64], happier mothers may have better resources to offer options to replace children’s screen time and encourage their children to engage in other activities. Alternatively, happier mothers are intrinsically motivated to practice healthy ERBs themselves, leading them to be role models for their children [65]. Also, happier mothers are more likely to practice warm parenting and interact with their children positively [66], which leads children to better comply with their mothers’ suggestions. 

### 4.2. Parental Play Engagement and Children’s ERBs 

Out of parental happiness, play engagement, and parenting stress, it was the parental play engagement that had the strongest association with child ERBs. Thus, our study underscores the crucial role of parents’ active participation in child play. This engagement not only provides a unique opportunity for parents to offer constructive support for their child’s play but also facilitates the development of a lasting and meaningful relationship between parents and children [67]. The close parent–child bond cultivated through play engagement further enhances parent–child attachment and bonding [67], fostering a positive and nurturing home environment. Additionally, the positive interactions during playtime enhance communication between parents and children, becoming a valuable tool for parents to guide their children toward constructive ERBs. In essence, shared activities, such as play, create a foundation where children are more receptive to parental guidance, promoting cooperation and responsiveness [68].

Another pathway from parental active play engagement to children’s ERBs may be that play stimulates children’s capacity for self-regulation. Self-regulation is an intricate construct that encompasses the capacity to manage behaviors, emotions, and cognitions in the face of environmental demands [69]. Parents who utilize such positive parenting are more likely to provide clear and consistent standards and boundaries for child behavior [70]. In addition, when parents consistently respond to their children’s cues and needs with sensitivity, warmth, and appropriateness, children establish emotional security [70]. This positive cycle fosters the development of strong self-regulation skills [71], which would enable them to engage in positive ERBs effectively. 

### 4.3. Maternal Parenting Stress and Child Healthy Food Consumption 

The association between mothers’ increased parenting stress and children’s decreased healthy eating is in line with previous findings. For instance, higher parenting stress was related to less fruit and vegetable intake by children [29]. Maternal parenting stress was found to reduce mothers’ motivation to stock healthy foods at home [22]. Another possible mechanism is that stressed mothers are more likely to engage in emotional eating and ingest unhealthy food frequently [72], thereby potentially serving as a negative role model for their children. 

### 4.4. Child Screen Time and Child Happiness

When we consider all four ERBs simultaneously in our study, child screen time emerges as the standout ERB with a consistent connection to both happiness and weight status lasting for at least two years. Previous research has consistently reported that high screen time is associated with various negative well-being and health outcomes [73]. For example, in a recent study in Spain with older children, increased screen time was related to poorer psychological well-being and greater psychological distress [74]. Also, adolescents and young adults with longer screen time reported lower psychological well-being [75], less happiness [76], and more stress [77]. Here, we showed that these negative side effects of increased screen time start already at the early elementary school age and are manifested in a different culture. 

Several mechanisms could explain the negative link between screen time and childhood happiness. Excessive screen time can lead to social isolation as individuals may spend less time interacting with others in person [78]. Prolonged screen time can reduce face-to-fact interactions with family and friends, which are crucial for emotional well-being and support [79]. Loneliness and social isolation are known risk factors for psychological distress [80]. Also, content on screens can vary widely, and exposure to distressing or violent content can increase feelings of fear, anxiety, or distress, further impacting a child’s emotional well-being [79,81]. These factors collectively underscore the potential for screen time to adversely affect a child’s happiness.

### 4.5. Child Screen Time and Child Weight Status

With the development of media-related technologies, a considerable amount of time is now being spent in front of a screen. Children who are exposed to screens for long periods may exacerbate the risk of overweight BMI and obesity due to a lack of PA and the tendency to ingest more high-calorie food [73]. Longer screen time is often accompanied by lower overall PA, being replaced with increased sedentary behavior, which would lead to lower energy expenditure and increased fat deposits and BMI [82]. Moreover, eating in front of a screen would delay hunger cues and lead to excessive food intake [83]. Therefore, longer screen time may be one of the most important risk factors for overweight BMI and obesity [14]. 

### 4.6. Parent’s and Child’s Gender 

We observed a few gender associations between parents and children in that maternal play engagement was negatively linked with boys’ unhealthy food consumption, maternal play engagement was positively related to girls’ PA, maternal happiness was negatively associated with girls’ screen time, and paternal happiness was positively related to boys’ unhealthy food intake. All in all, it was the mothers that had a positive influence on a child’s health and development. This may be attributed to the gender differences in traditional parental roles. Mothers predominantly assume the proactive role in child rearing, planning, and household management [84,85]. In contrast, fathers, often lacking expertise in housework or childcare, tend to be less engaged [86]. As mothers take on more responsibility for feeding the child, especially in a healthy and nutritious way [86,87], fathers may resort to easy but less nutritious meals and snacks when they are in charge, possibly due to their limited cooking skills or lower level of attentiveness.

According to a recent study in Korea that aimed to explore the division of childcare responsibilities among parents, mothers assumed 70.9% of the childcare duties during weekdays regardless of employment status [88]. During weekends, fathers’ involvement increases, but still mothers bear more responsibility at 57.8% [88]. Nevertheless, a shift is noticeable as many fathers are now demonstrating willingness to actively engage in family caregiving. They are increasingly taking parental leave from work and becoming more involved in child rearing [89]. These findings shed light on the existing gender disparities in parenting roles and emphasize the need for further research and societal efforts to promote gender equality in childcare responsibilities.

### 4.7. Limitations 

Despite several novel findings of this study, the results should be considered in light of limitations. First, this is an observational study from which causation cannot be determined. Second, the sample is homogeneous Korean married parents and their children, who are developing in the broad normal range. Moreover, some of the assessments reflect Korean culture, such as the type of play addressed when assessing parental play engagement and marker foods targeted when measuring a child’s intake. These findings need to be replicated with other samples while still needing to reflect cultural competencies. Third, child ERBs were reported mostly by mothers, which could be biased toward presenting a more positive picture than reality. Employing more objective assessment tools such as tracking devices or ecological momentary assessment via smartphone applications could enhance accuracy and reliability. Fourth, our understanding of screen behaviors is limited to the amount of time spent in front of a screen, as reported by a parent. Therefore, we are unable to examine the content or context of the screen time that may moderate child ERBs. For example, future research should investigate whether certain types of media or content exposure are linked to more harmful outcomes. Finally, acknowledging the potential impact of parental factors on a child’s self-report adds a layer of complexity to the interpretation of happiness measurements in children. Consequently, further research is necessary to evaluate the validity of children’s self-reported happiness. Notwithstanding these limitations, this is the first study that illuminates the potential mechanisms of how various parental factors may influence Korean children’s ERBs concurrently and ultimately their happiness and weight status over time.

## 5. Conclusions

Healthy and well-adjusted child development is an overriding goal of most parents as well as society [90]. In our pursuit of identifying the optimal approach to enhance children’s happiness and promote healthy weight outcomes, we found that broader parenting factors can serve as protective factors for both childhood happiness and weight status among Korean 7-to-9-year-olds. This association is linked to a decrease in children’s screen time. Particularly, we could argue that when parents actively play with them, expressing attention and care, children may unwittingly come to utilize self-regulation to practice healthier behaviors, but further studies are required to understand the possible mechanisms. Furthermore, the unexpected finding regarding the paternal role in boys’ unhealthy food intake underscores the need for societal support to encourage greater paternal involvement in their children’s lives. This could involve initiatives promoting a work–life balance and public health campaigns offering practical suggestions on how fathers can engage more actively with their children. Finally, in today’s digital age, where children have access to a wide range of media for both entertainment and education, there is an urgent need for a personalized approach to screen content and duration. In this sense, it becomes crucial to advocate for alternative leisure activities within families and equip parents with resources to facilitate and encourage these options.

## Figures and Tables

**Figure 1 ijerph-21-00176-f001:**
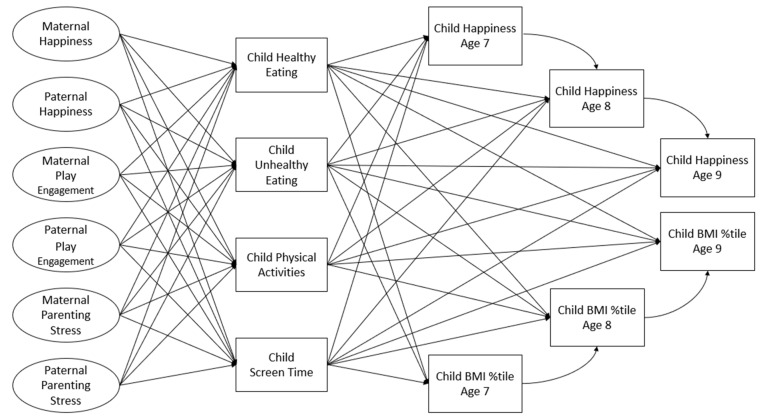
Conceptual model.

**Figure 2 ijerph-21-00176-f002:**
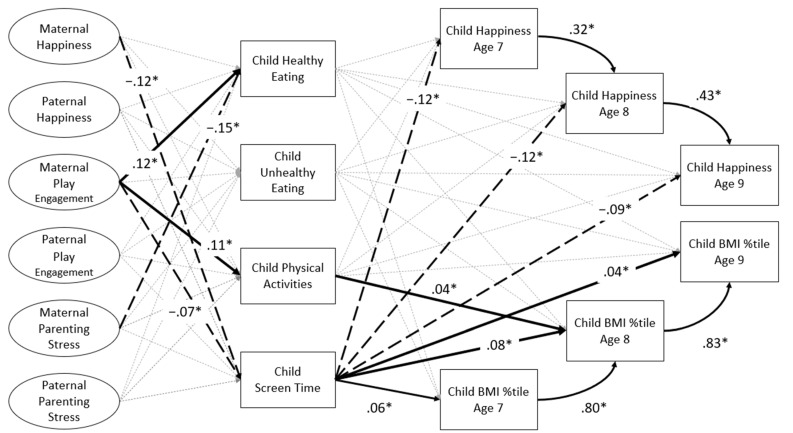
Path analysis results for the total sample. Latent variables are represented with ovals and observed variables by rectangles. All parental variables and child healthy eating, unhealthy eating, physical activities, and screen time were assessed at child age 7. Only significant standardized path coefficients are reported. Positive relations are represented by bold lines; negative relations are represented by dotted lines. All paths were controlled for mothers’ and fathers’ education level, and household composition. BMI %tile = body mass index percentile. * *p* < .05.

**Figure 3 ijerph-21-00176-f003:**
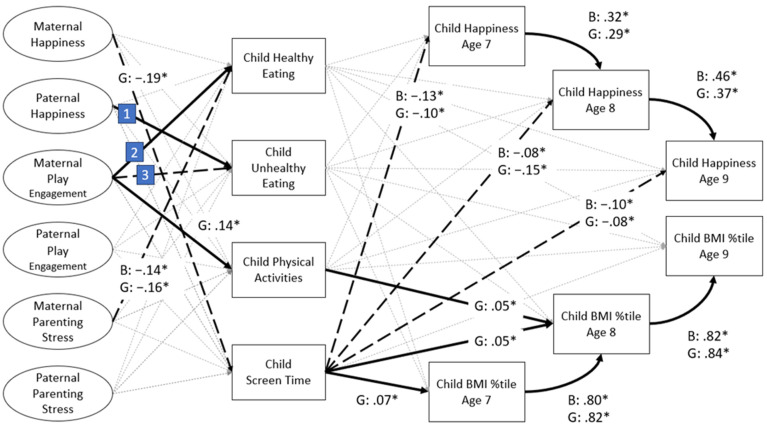
Path analysis results for the multi-group analysis. Latent variables are represented with ovals and observed variables by rectangles. All parental variables and child healthy eating, unhealthy eating, physical activities, and screen time were assessed at child age 7. Only significant standardized path coefficients are reported. Positive relations are represented by bold lines; negative relations are represented by dotted lines. All paths were controlled for mothers’ and fathers’ education level, and household composition. BMI %tile = body mass index percentile; B = Boys; G = Girls. [1] B: .13 *; [2] B: .12 *; G: .11 *; [3] B: −.10 *; * *p* < .05.

**Table 2 ijerph-21-00176-t002:** Descriptive Statistics for Parental Variables.

Parental Variables		Scale	Total	Mother	Father
			M (SD)	M (SD)	M (SD)
Happiness					
	In general, I consider myself happy.	1–7	5.40 (1.08)	5. 36 (1.10)	5.44 (1.07)
	Compared with most of my peers, I consider myself happy.		5.32 (1.11)	5.27 (1.14)	5.37 (1.08)
	I enjoy life regardless of what is going on.		5.03 (1.18)	5.02 (1.22)	5.05 (1.13)
Play Engagement					
	I do arts and crafts with my child.	1–4	1.90 (0.71)	2.05 (0.70)	1.74 (0.68)
	I do puzzles and/or board games with my child.		1.90 (0.69)	1.90 (0.70)	1.90 (0.69)
	I build things and/or play with blocks with my child.		1.83 (0.68)	1.91 (0.71)	1.74 (0.66)
	I talk about nature or do STEM projects with my child.		1.78 (0.68)	1.77 (0.71)	1.78 (0.66)
Parenting Stress					
	I am not sure if I could be a good parent.	1–5	2.30 (0.88)	2.42 (0.90)	2.18 (0.84)
	I am not sure if I could raise my child properly.		2.15 (0.85)	2.26 (0.97)	2.03 (0.79)
	I feel sometimes that my child is behind his or her peers because I am not doing enough as a parent.		2.23 (0.94)	2.33 (0.84)	2.22 (0.88)

Note. SD = standard deviation.

**Table 3 ijerph-21-00176-t003:** Descriptive Statistics for Child Variables.

Child Variables			Child, M (SD)	Boys, M (SD)	Girls, M (SD)	Boys vs. Girls *t*-Test (*df* = 1526)
	Healthy eating	4–12	8.74 (1.76)	8.78 (1.84)	8.69 (1.68)	t = 1.03, *p* = 0.02
	Unhealthy eating	3–9	6.39 (1.56)	6.48 (1.61)	6.30 (1.50)	t = 2.24, *p* = 0.01
	Physical activities (min/week)	0–56	27.03 (6.50)	28.02 (6.00)	25.98 (6.83)	t = 2.34, *p* = 0.13
	Screen time (h/week)	0–50	13.04 (7.07)	13.45 (7.17)	12.60 (6.95)	t = 5.26, *p* = 0.08
	Child happiness at age 7	0–24	19.52 (2.85)	19.09 (2.91)	19.99 (2.71)	t = −6.31, *p* = 0.01
	Child happiness at age 8	0–24	19.98 (2.63)	19.52 (2.82)	20.45 (2.32)	t = −6.84, *p* = 0.01
	Child happiness at age 9	0–24	19.92 (2.71)	19.60 (2.92)	20.26 (2.43)	t = −4.58, *p* = 0.01
	BMI percentile at age 7	1–100	53.43 (31.30)	53.56 (31.46)	53.31 (31.15)	t = 0.17, *p* = 0.56
	BMI percentile at age 8	1–100	58.40 (31.30)	62.33 (30.87)	54.31 (31.25)	t = 4.86, *p* = 0.71
	BMI percentile at age 9	1–100	55.76 (30.52)	58.90 (30.43)	52.54 (30.32)	t = 3.90, *p* = 0.23

Note. SD = standard deviation.

**Table 4 ijerph-21-00176-t004:** Factor Loadings and Model Fit for Indicators of Latent Variable.

Measurement Models		Mother		Father	
		Factor Loadings	Model Fit	Factor Loadings	Model Fit
Parental Happiness					
	In general, I consider myself happy.	0.93	RMSEA = 0.00 CFI = 1.00 TLI = 1.00	0.93	RMSEA = 0.00 CFI = 1.00 TLI = 1.00
	Compared with most of my peers, I consider myself happy.	0.94		0.94	
	I enjoy life regardless of what is going on.	0.78		0.78	
Play Engagement					
	I do arts and crafts with my child.	0.69	RMSEA = 0.06 CFI = 0.99 TLI = 0.98	0.69	RMSEA = 0.05 CFI = 1.00 TLI = 0.99
	I do puzzles and/or board games with my child.	0.76		0.76	
	I build things and/or play with blocks with my child.	0.75		0.75	
	I talk about nature or do STEM projects with my child.	0.76		0.76	
Parenting Stress					
	I am not sure if I could be a good parent.	0.91	RMSEA = 0.00 CFI = 1.00 TLI = 1.00	0.88	RMSEA = 0.00 CFI = 1.00 TLI = 1.00
	I am not sure if I could raise my child properly.	0.93		0.91	
	I feel sometimes that my child is behind his or her peers because I am not doing enough as a parent.	0.74		0.73	

Note. RMSEA = Root Mean Square Error of Approximation, CFI = Comparative Fit Index, TLI = Tucker Lewis Index.

## Data Availability

The data used in this study are available to individual researchers or institutions upon approval of the Korea Institute of Child Care and Education (KICCE, https://kicce.re.kr, accessed on 21 April 2023). The original contributions presented in this study can be found in Mendeley Data, V1, https://doi.org/10.17632/82ckn4ptzm.1.

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
