# Peer review of "Associations of Broader Parental Factors with Children’s Happiness and Weight Status through Child Food Intake, Physical Activity, and Screen Time: A Longitudinal Modeling Analysis of South Korean Families"

_ijerph, 2024, doi:10.3390/ijerph21020176_

Round 1

Reviewer 1 Report

Comments and Suggestions for Authors

The author’s aimed to determine parenting factors associated with children’s energy-related behaviors (ERBs) including eating, physical activity, and screen time. They also determined the role of energy-related behaviors on the children’s BMI and happiness. They had 4 hypotheses: 

1)     parental happiness is associated with more healthy and less unhealthy ERBs in their children.

2)     parent-child play can provide a practical avenue for enhancing the parent-child relationship.

3)     parenting stress will be associated with fewer positive and more negative ERBs among children.

4)     the mothers’ influence on children’s ERB will be stronger than those of fathers.

They report that maternal play engagement rather than maternal happiness was associated with the most ERBs, screen time was the only ERB that was associated with the children’s happiness and BMI at all 3 ages and that the mother’s happiness, play engagement and stress was associated with the children’s ERBs more then the father’s variables. While the manuscript was clear, a few questions remain.

1.     Table 3. define which comparisons were used for the p-value

2.     Table 1-3. Are the demographics as well as the descriptive stats for parents (table 2) and children (table 3) representative and comparable to prior waves of the PSKC?  

3.     Table 4 (correlation table). The rationale for selecting these specific questions and reporting the associations was unclear. Also, section 3.1 is lacking a description of any of the correlations, which further questions the relevance of the associations reported.  Are there specific correlations that the author’s find particularly important, relevant or unexpected, etc.? Adding a description would support their inclusion in the manuscript. Finally, it is very difficult to both read and interpret the table since the text is too small and the reader must compare the numbers to the key for the responses. By using color to represent both the direction of the correlation and significant correlations, the very small symbols could be eliminated and the reader would be focused on significant correlations.

4.     Section 3.4 and Figure 3. It was unclear why the paths between maternal parenting stress and healthy eating for both boys and girls was excluded from the text since path coefficients were included in the figure.

5.     Sections 3.3 to 3.4 and Figures 2 and 3. It was unclear why the path coefficients for child happiness between ages as well as BMI percentile between the ages were excluded from the text. 

6.     Section 4.2 in the discussion. This section focused on parent’s active participation. Yet, the results show now no associations between the paternal variables and ERBs. Are there other studies that also support the results? 

7.     The manuscript noted the greater division of childcare responsibilities among parents regardless of the day of the week. Thus, the lack of associations between paternal variables and ERBs are likely due to the 

Comments on the Quality of English Language

minor spelling and grammer errors.

Reviewer 2 Report

Comments and Suggestions for Authors

The research is interesting and relevant, the methodological component of the work corresponds to the stated tasks. At the same time, the terminology used by the authors (Broader Parental Factors or energy-related behaviors, for example), presented as well-known, but not particularly justified in the text, raises questions.

The question of the reliability of the selected psychological tests is also interesting, because we are talking about evaluating generally rather "abstract" concepts such as happiness (and in the study it is evaluated using the same questionnaire for both adults and children - although it is adapted for the latter).  All this raises certain doubts about the seriousness of the study and the conclusions and conclusions obtained on the basis of its results and requires a more serious scientific examination, including using modern literary data on this subject.

Round 2

Reviewer 2 Report

Comments and Suggestions for Authors

The authors have done a lot of work and significantly revised the text of the article.

The methodology for assessing the level of happiness in children still raises questions - it is unclear whether the results obtained are really reliable, since initially the same questionnaire is used as for adults in a revised form. It is unclear to what extent the children involved in the study really understood the meaning of the "faces" with which the authors replaced the answer options in the adult questionnaire (it is also unclear, in fact, what the correspondence between them is). The factor of influence of parental opinion or possible guidance to the child during such a survey is not taken into account.

The conclusions drawn are very theoretical in nature and do not seem to correspond to the tasks set.

In general, the work and its methodology raise certain doubts about the seriousness of the conducted research and the conclusions obtained on the basis of its results, and the conclusion requires a more serious scientific justification.

In the opinion of the reviewer, the work does not meet the requirements for such scientific publications.
